# Cytotoxicity of *Mimusops caffra*-Based Ursolic Acid, Oleanolic Acid and Derivatives Against Human Cancerous and Non-Cancerous Cell Lines

**DOI:** 10.3390/ijms26209969

**Published:** 2025-10-13

**Authors:** Sithenkosi Mlala, Opeoluwa Oyehan Oyedeji, Gbemisola Morounke Saibu, Mavuto Gondwe, Adebola Omowunmi Oyedeji

**Affiliations:** 1Department of Chemistry, Faculty of Science and Agriculture, University of Fort Hare, Private Bag X1314, Alice 5700, South Africa; stherah18@gmail.com; 2Department of Chemical and Physical Sciences, Faculty of Natural Sciences, Walter Sisulu University, Private Bag X1, Mthatha 5117, South Africa; morounkesaibu@gmail.com; 3Department of Human Biology, Faculty of Health Sciences, Walter Sisulu University, Private Bag X1, Mthatha 5117, South Africa; mgondwe@wsu.ac.za

**Keywords:** cancer, ursolic acid, 3-O-acetyl ursolic acid, ursolic-28-methylate, 3-acetylursolic-methylate, cytotoxicity, MDA, KMST-6, HepG2, PC3

## Abstract

According to the World Health Organization, cancer is still the leading cause of death for humans worldwide. Although over 100 chemotherapeutic agents are currently available for the treatment of cancer patients, the overall long term clinical benefit is disappointing due to the lack of effectiveness or severe side effects from these drugs. The use of complementary and alternative medicinal products from plants has continued to increase in past decades, due to fewer side effects of bioactive compounds from medicinal plants of which pentacyclic triterpenoids have been identified as one class of secondary metabolites that could play an important role in the treatment and management of a number of non-communicable diseases. The main aim of this study is to extract, isolate, identify, and elucidate pentacyclic triterpenoid (ursolic acid, UA (**1**), and oleanolic acid, OA (**2**)) *from Mimusops caffra.* Semi-synthesis of UA was carried out to obtain some triterpenoid derivatives (3-O-acetyl ursolic acid, AUA (**3**), ursolic-28-methylate, UM (**4**), and 3-acetylursolic-methylate, AUM (**5**)), and we evaluated these compounds as anti-cancer therapeutic agents. Isolation of ursolic acid (UA) (**1**) from *M. caffra* is always accompanied by its isomer oleanolic acid (OA) (**2**) due to their similar retention factors (Rf) values. Acetylation and deacetylation techniques were used to isolate compounds **1** and **2**. In vitro cytotoxicity activities of UA, AUA UM, and AUM were evaluated against various cancer cell lines, such as human breast adenocarcinoma cancer cell lines (MDA), human liver cancer cell lines (HepG2), human prostate cancer cell lines (PC3) and non-cancerous human fibroblast cell lines (KMST-6) using MTT assays. The UM exhibited remarkable cytotoxic activities against cancer cells, while little or no activities were observed on non-cancerous cell lines, which indicates that the addition of methyl at C-28 of UA is essential to enhance its activity as a therapeutic agent for cancer. The AUA showed moderate or no cytotoxicity against the different cancer cell lines, which is less than that of the UA parent compound. Moreover, these results suggest that ursolic acid and UA derivatives are potential therapeutic drugs for human breast, liver, and prostate cancers.

## 1. Introduction

Noncommunicable diseases (NCDs) are still the world’s leading cause of death, accounting for 43 million deaths in 2021, of which 10 million deaths were caused by cancer. Cancer is the second-largest cause of premature death in 134 of the world’s 183 countries [1]. Although tuberculosis, influenza and pneumonia, HIV, cerebrovascular diseases, and diabetes mellitus are the leading causes of death, cancer and other NCDs account for 4.6% of the world’s deaths.

Breast cancer is the most prevalent cancer among women worldwide. Of the 1 million women worldwide diagnosed with breast cancer, 140,000 die each year. Consequently, breast cancer is a diverse disease that requires different treatment strategies [2,3]. Since most cancer chemotherapeutics severely affect the normal cells of the host, the use of natural products in cancer control has now been explored with considerable value. Furthermore, finding new sources of biologically active compounds is important for the discovery of new drugs for cancer treatment [4]. Recent anticancer medications used for chemotherapy are harmful to normal cells, namely, they cause immune cell toxicity. Therefore, reducing doses to the lowest amount and reducing the side effects of those medications is important. Developing novel anticancer drugs with little or no side effects on the immune system is vital [5].

Prostate cancer is the second-most diagnosed disease in males worldwide, and the fifth-leading cause of cancer-related deaths [6]. Distant metastasis is the primary cause of tumor-associated mortality in prostate cancer. Through traditional imaging methods, the distribution of tumor cells from the primary tumor to distant locations through the bloodstream cannot be detected early. Circulating tumor cells are an important prognostic and predictive biomarker, capable of tracking the efficacy of adjuvant therapies, detecting early metastasis growth, and ultimately, evaluating advanced disease therapeutic responses earlier than conventional diagnostic methods [7]. Although over 100 chemotherapeutic agents are currently available for the treatment of cancer patients, the overall long-term clinical benefit is disappointing due to the lack of effectiveness or severe side effects from these agents [8].

Liver damage is among the most dangerous diseases that have been caused by the introduction of modern food styles and exposure to many toxins in the air, and heavy medication consumption [9]. The liver is known to be the chief site for intense metabolic activity and the secretion of several salts and pigments that aid proper digestion. It is one of the main and most vital organs in the human body. It assists in properly maintaining, supporting, and controlling the homeostasis in the human body [10]. Many in vitro and in vivo studies have indicated that oxidative stress and damage are associated with different types of liver disease [11]. With very few treatment options to counter its prevalence, liver cancer remains the world’s third-highest cause of cancer deaths.

Natural compounds derived from various sources, such as plants, fruits, vegetables, herbs, and even fungi, open a novel and exciting perspective on the treatment and prevention of hepatic cancer. Terpenoids have emerged as a promising group of phytochemicals amongst the various secondary metabolites produced by plants and other species. Terpenoids selectively kill cells of hepatic cancer with a pleiotropic mode of action while saving normal cells [12].

Of the vast number of 250,000–500,000 species of world flora, only 1–10% of plants have been scientifically assessed. There is a significant gap between the natural flora strength and the human ability to manipulate biodiversity with minimal or no adverse reactions for developing drugs, especially against cancer. Plant-derived medicinal constituents are not only essential as therapeutic agents, but they could also serve as templates for the synthesis of effective drugs with desired therapeutic profiles [13]. In cytotoxic assessments, a compound is regarded as significantly active when IC_50_ is lower than 30 μg/mL [14].

*Mimusops caffra* (Sapotaceae), commonly referred to as the coastal red milkwood, is an ecologically important tree in South Africa [15]. *M. caffra* is an indigenous small to medium-sized evergreen tree that grows in large, uniform clusters in the dunes of the Eastern Cape, KwaZulu-Natal coast, Southern Africa, and Mozambique [16]. The tree usually grows to a height of about 10 m and sometimes reaches a height of 25 m. The color of the bark is dark grey and contains milky latex. The leaves are hardy and leathery, with an upper blue-green color and have whitish hairs below [16,17]. It performs significant ecological functions, such as stabilizing coastal forest sands. The fruit can be used for various food items, and some plant sections are used in traditional medicine. The bark is utilized in herbal medicine to treat wounds and bruises. The plant produces berry-like, soft, pulpy fruits, which are eaten by local South Africans and are also used in the manufacture of jelly and alcohol [15,16]. Researchers in search of biologically active compounds from plants have investigated *M. caffra* against several diseases. For example, Simelane et al. [18] isolated and semi-synthesized a few compounds from *M. caffra* with interesting anti-plasmodial activity. The crude extract of *M. caffra* leaf material displayed the strongest activity, with an IC_50_ of 2.14 μg/mL. The pentacyclic triterpenoid ursolic acid, isolated from the leaves of *M. caffra* was the most active compound (IC_50_ =6.8 μg/mL) in contrast to taraxerol and sawamilletin isolated from *M. obtusifoli* (IC_50_ > 100). Chemical modification of ursolic acid to 3β-acetylursolic acid significantly increased its anti-plasmodial efficacy. According to in vitro experiments in mice, the 3β-acetylursolic acid decreased parasitemia against *Plasmodium berghei* by 94.01%. The 3β-acetylursolic acid cytotoxicity IC_50_ of two human cell lines (HEK293 and HepG2) was 366.00 μg/mL and 566.09 μg/mL, respectively. The findings confirm folk medicine use of these plants [18].

Many plant-based terpenoids display promising therapeutic activities, but their solubility, stability, and bioavailability are still a challenge for their clinical use. The semi-synthesis and synthesis of these compounds enhances their biological activity, solubility, stability, and bioavailability [19,20,21]. Therefore, an investigation of the effects of the plant-derived compounds and their derivatives as therapeutic agents would assist and improve the quality of modern drug discovery.

## 2. Results

The results describing the structure–activity relationship for *M. caffra*-based pentacyclic triterpenes, such as ursolic acid, oleanolic acid, and UA derivatives, are presented in the hope of developing new anticancer drugs. From the 1 kg of *M. caffra* used, UA gave 92% yield, while OA was 82%. The spectroscopic spectra of the 5 compounds are presented in Appendix A Figure A1, Figure A2, Figure A3, Figure A4, Figure A5, Figure A6, Figure A7, Figure A8, Figure A9, Figure A10, Figure A11, Figure A12, Figure A13, Figure A14, Figure A15, Figure A16, Figure A17, Figure A18, Figure A19, Figure A20, Figure A21, Figure A22, Figure A23, Figure A24, Figure A25, Figure A26, Figure A27, Figure A28, Figure A29, Figure A30 and Figure A31.

### 2.1. Spectroscopic Analysis of Isolated Compounds and Derivatives

#### 2.1.1. Ursolic Acid (**1**)

A white crystalline powdered ursolic acid (C_30_H_48_O_3_, UA) was observed to have the following chemical properties: percentage yield of 92%, Mz+ of 456.8, and Mp of 285–286 °C. FTIR (ATR, *v*_max_ cm^−1^) showing strong absorption bands of 3386 cm^−1^ (broad free alcohol -OH), 2852, 2935 cm^−1^ (aliphatic sp3 –C-H), 2871 cm^−1^ (aromatic sp2-C=C), and 1688 cm^−1^ (carboxylic acid -C=O). ^1^H-NMR (600 MHz, DMSO) and ^13^C DEPT: (3H, 7 methyl protons namely C23–C27 and C29–C30) of 1.11 s, 1.11 s, 1.16 s, 1.16 s, 1.26 s, 1.06 d, 1.06 s, respectively; (2H, 18 methylene protons of C1, C2, C6, C7, C15, C16, C21 and C22) of 1.49 α, 1.24 β, 1.72 α, 1.47 β, 1.52 α, 1.27 β, 1.49 α, 1.24 β, 2.05 α, 1.80 β, 1.38 α, 1.13 β, 1.61 α, 1.56 β, 1.52 α, 1.27 β, 1.75 α, 1.50 β, respectively and (^1^H, 7 methine protons of C3, C5, C9, C¬12, C18, C19 and C20) of 4.34, 1.39, 1.43, 5.29, 2.62, 1.63 and 1.60. ^13^C-NMR (600 MHz, DMSO, C1–C30): 33.2, 27.4, 77.3, 40.5, 55.2, 21.6, 36.8, 39.9, 47.5, 37.8, 24.3, 122.3, 143.9, 42.1, 28.7, 27.4, 45.9, 52.8, 38.9, 38.8, 28.7, 32.5, 18.5, 18.5, 21.6, 17.0, 21.4, 178.8, 16.6 and 17.0. The structure elucidation was further confirmed by 2-dimensional ^13^C-NMR including COSY, HMBC, and HSQC.

#### 2.1.2. Oleanolic Acid (**2**)

A white amorphous powdered oleanolic acid (C_30_H_48_O_3_, OA) was reported to have the following chemical properties: percentage yield of 82%, Mz+ of 456.4, and Mp of 297–299 °C. The structural elucidation of FTIR (ATR, *v*_max_ cm^−1^) of OA confirmed the major peaks at 3406 (free alcohol stretch–OH), 2835, 2864 (aliphatic sp3 stretch –C-H), 1688 (carboxylic acid -C=O band), and 1460 (aromatic sp3 -C=C). 1H-NMR (400 MHz, CDCl_3_): (3H, 7 methyl protons namely C23–C27 and C29–C30) of 1.12 s, 1.12 s, 1.15 s, 1.15 s, 1.28 s, 1.10 s, 1.10 s, respectively; (2H, 18 methylene protons of C1, C2, C6, C7, C15, C16, C19, C21 and C22) of 1.45 α, 1.24 β, 1.72 α, 1.48 β, 1.52 α, 1.28 β, 1.45 α, 1.24 β, 2.11 α, 1.80 β, 1.38 α, 1.15 β, 1.61 α, 1.55 β, 2.73 α, 2.77 β, 1.45 α, 1.24 β, 1.73 α, 1.51 β, respectively and (1H, 7 methine protons of C3, C5, C9, C¬12 and C18) of 3.16, 1.38, 1.45, 5.17 and 2.74. 13 C-NMR (400 MHz, CDCl_3_, C1–C30): 35.3, 27.1, 79.0, 41.0, 55.2, 23.4, 37.1, 39.3, 47.6, 38.4, 26.0, 122.5, 143.7, 45.9, 30.7, 25.6, 47.6, 41.6, 45.5, 32.6, 41.0, 32.5, 18.3, 18.3, 23.4, 17.1, 23.6, 183.0, 27.7 and 27.7. The structure of OA was further elucidated by 2-dimensional ^13^C-NMR, including COSY, HMBC, and HSQC.

#### 2.1.3. Acetyl Ursolic Acid (**3**)

A white powdered 3β-acetyl ursolic acid (C_32_H_50_O_4_, AUA) was confirmed to have the following chemical structural properties: percentage yield of 94%, Mz+ of 499.8, and Mp of 288–289 °C. FTIR (ATR, *v*_max_ cm^−1^) major absorption bands, confirm the formation of an acetyl group in position C3 with the presence of a broad free alcohol (OH) peak at 3349 cm^−1^, sp3 C-H bands at 2935 and 2851 cm^−1^, the carboxylic (C=O) at 1731 and 1689 cm^−1^ with C12-C13 occupied by C=C at 1466 cm^−1^ and C-O at 1047 cm^−1^. 1 H-NMR (400 MHz, CDCl_3_): (3H, 8 methyl protons namely C23–C27, C29–C30 and C32) of 1.09 s, 1.09 s, 1.17 s, 1.17 s, 1.26 s, 1.05 d, 1.05 s and 2.07 s, respectively; (2H, 18 methylene protons of C1, C2, C6, C7, C15, C16, C21 and C22) of 1.48 α, 1.26 β, 1.77 α, 1.55 β, 1.50 α, 1.26 β, 1.48 α, 1.26 β, 2.07 α, 1.80 β, 1.39 α, 1.14 β, 1.60 α, 1.35 β, 1.54 α, 1.26 β, 1.75 α, 1.50 β, respectively and (1H, 7 methine protons of C3, C5, C9, C¬12, C18, C19 and C20) of 4.36, 1.39, 1.43, 5.15, 2.48, 1.64 and 1.58. 13 C-NMR (400 MHz, CDCl_3_, C1–C30 and C32): 33.1, 23.5, 80.9, 37.6, 55.2, 18.1, 35.8, 39.9, 47.5, 37.6, 25.8, 121.9, 144.0, 41.6, 30.6, 28.0, 45.1, 55.1, 39.4, 39.2, 28.0, 32.5, 18.1, 18.1, 23.2, 17.0, 21.2, 180.3, 16.7, 17.0 and 21.2.

#### 2.1.4. Ursolic-28-Methylate (**4**)

A white powdered methyl ursolate (C_31_H_50_O_3_, UM) was observed to have a percentage yield of 98%, Mz+ of 469.5, and Mp of 141–143 °C. FTIR (ATR, *v*_max_ cm^−1^) showing strong absorption bands of free alcohol, OH (3431 cm^−1^), C-H stretches (2941 and 2861 cm^−1^), carboxylic C=O (1725 and 1690 cm^−1^), C=C (1468 cm^−1^), and C-O (1024 cm^−1^). 1 H-NMR (400 MHz, CDCl3): (3H, 8 methyl protons namely C23–C27, C29–C30 and C31) of 1.11 s, 1.11 s, 1.14 s, 1.14 s, 1.23 s, 1.07 d, 1.07 s, 3.53 s, respectively; (2H, 18 methylene protons of C1, C2, C6, C7, C15, C16, C21 and C22) of 1.48 α, 1.25 β, 1.71 α, 1.46 β, 1.57 α, 1.23 β, 1.48 α, 1.25 β, 1.32 α, 1.11 β, 1.82 α, 1.39 β, 1.57 α, 1.25 β, 1.82 α and 1.65 β, respectively and (1H, 7 methine protons of C3, C5, C9, C¬12, C18, C19 and C20) of 3.00, 1.45, 1.46, 5.17, 2.78, 1.65 and 1.61. 13 C-NMR (600 MHz, DMSO, C1–C30): 33.6, 27.4, 77.3, 41.3, 53.0, 21.4, 37.0, 40.0, 47.5, 38.5, 26.1, 122.3, 143.9, 41.6, 30.8, 27.6, 46.4, 55.2, 39.3, 38.8, 28.7, 32.5, 18.5, 18.5, 23.4, 17.0, 21.4, 177.5, 16.5, 18.5 and 51.9.

#### 2.1.5. 3-Acetylursolic-Methylate (**5**)

A white crystalline powder, 3-acetylursolic-methylate (C_33_H_52_O_4_ AUM), was observed with a percentage yield of 77%, Mz+ of 516.0, and Mp of 227–228 °C. FTIR (ATR, vmax cm^−1^) shows strong absorption bands C-H stretches (2937 and 2861 cm^−1^), carboxylic C=O (1734 and 1694 cm^−1^), C=C (1466 cm^−1^), and C-O (1050 cm^−1^). 1 H-NMR (400 MHz, CDCl3): (3H, 9 methyl protons namely C23–C27, C29–C30, C31 and C33) of 1.00 s, 1.00 s, 1.18 s, 1.18 s, 1.25 s, 1.06 d, 1.06 s, 1,98 s, and 3.53 s, respectively; (2H, 16 methylene protons of C1, C2, C6, C7, C15, C16, C21 and C22) of 1.47 α, 1.25 β, 1.81 α, 1.56 β, 1.52 α, 1.27 β, 1.47 α, 1.25 β, 2,14 α, 1.79 β, 1.39 α, 1.06 β, 1.73 α, 1.47 β, 1.85 α and 1.65 β, respectively and (1H, 7 methine protons of C3, C5, C9, C¬12, C18, C19 and C20) of 4.41, 1.39, 1.43, 5.16, 4.41, 1.62 and 1.61. 13C-NMR (600 MHz, DMSO, C1–C30): 32.9, 24.2, 80.9, 37.7, 55.3, 21.2, 36.7, 39.5, 47.5, 37.7, 24.2, 125.5, 138.2, 42.0, 30.7, 28.0, 47.5, 51.5, 39.0, 38.9, 28.1, 32.9, 21.3, 21.3, 23.3, 17.1, 21.2, 178.1, 16.7, 16.9, 21.2, 171.0 and 52.9.

### 2.2. Cytotoxicity of Pentacyclic Triterpenoids and Derivatives

In this paper, *M. caffra* isolated compounds (**1**), and derivatives (**3**, **4** and **5**) were investigated as potential cancer therapeutic agents against breast adenocarcinoma cancer cell lines (MDA), human liver cancer cell lines (HepG2) and human prostate cancer cell lines (PC3), and non-cancerous human fibroblast cell lines (KMST-6) using MTT assay. The calculated % cytotoxicity (Figure 1a–d) was used to determine the cell proliferation towards the human cell lines from each compound and extrapolate the IC_50_ values (Table 1).

## 3. Discussion

Figure 1a presents a summary of the cytotoxicity activity of isolated compounds such as ursolic acid (**1**) with semi-synthesized derivatives such as 3-O-acetyl ursolic acid (**3**) and methyl ursolate (**4**) and 3-acetylursolic-methylate (**5**) against MDA, KMST-6, HepG2, and PC3 cell lines. In the present study, the addition of a methyl group in C-28 of ursolic acid (UA) to yield methyl ursolate (UM) showed the highest potency at IC50 = 2.48 µM (*p* < 0.0001) against MDA compared to the ursolic acid (UA) parent compound with an IC_50_ value of >5 µM (*p* < 0.05). UA modifications have focused mainly on its functional groups at position C-3 (OH) and position C-28 (COOH). The addition of polar groups or active groups into the main structure significantly improves the efficacy of UA derivatives against cancer and enhances water solubility [22]. One of the main characteristics is the presence of an intact carbonyl group at position C-28; reduction or elimination of this moiety has led to compounds with low cytotoxicity [23].

Many potential biochemical effects of ursolic acid have been investigated, including antiproliferative, anti-inflammatory, and antioxidant capabilities. In vitro, ursolic acid prevents the proliferation of different types of cancer cells by inhibiting the signal transducer and transcription of the three activation pathways and can also decrease cancer cell proliferation and induce apoptosis [24]. Ursolic acid inhibits cell proliferation against MDA-MB-231 at a concentration ≥160 µg/mL [25]. In one study, ursolic acid demonstrated potent cytotoxic activity against HeLa, HT-29, and MCF-7 cells with IC_50_ values of 10, 10, and 20 μM, respectively [26]. UA also showed potential anti-cancer, anti-inflammatory, and antioxidant activity in several breast cancer cells in humans [3,27,28]. Previous studies also showed that UA can reduce proliferation and induce apoptosis in breast, prostate, lung, and endometrium leukemia, melanoma, and cancers. UA was shown to prevent tumor growth, induce differentiation of tumor cells, and demonstrate antiangiogenic activity [29]. The ursolic acid is currently used in human clinical trials to treat cancer, tumors, and skin wrinkles [30]. Because of its tumor origin, MDA-MB-231 has some interesting properties, such as high endogenous migratory capacity and the ability to sustain some degree of proliferation under serum-deprived culture. In the human breast cancer cells (MDA-MB-231) bioassay, OA clearly enhanced cell migration [31].

The esterification at the hydroxyl position C-3 by the acetyl group reduced the effect as its cytotoxicity activity was less than that of the UA parent compound. Gu and the co-authors [32] reported a sequence of UA derivatives of quinoline and oxadiazole. The cytotoxicity of synthesized derivatives against cancer cell lines such as human breast (MDA-MB-231) was evaluated by these researchers using the MTT assay. As a positive control, the cancer drug etoposide has been used. The findings of this study showed significant effects on at least one of the cancer cell lines (MDA-MB-231) (IC_50_ < 10 μM) of UA derivatives [32]. Ursolic acid and methyl ursolate were directly isolated from the hexane extract by biological activity-guided fractionation of *Mitracarpus frigidus*. Ursolic acid was observed to be effective in tumor cell lines such as human leukemia cell (HL60), Jurkat, MCF-7, and human colon cancer cells (HCT) with ED_50_ values ranging from 4.2 to 35.7 µg/mL, and methyl ursolate was only prominent in HL60 cells with an ED_50_ of 22.7 µg/mL [33]. Ma et al. [34] reported the good cytotoxicity activity of UA and derivatives including 3-O-acetyl ursolic acid and methyl ursolate against human leukemia cancer cell line (HL-60); human gastric cancer cell line (BGC-823); human breast cancer cell line (MDA-MB-435); human cervical cell line (Hela), and human hepatocellular carcinoma cell line (Bel-7402) [34].

Many in vitro and in vivo studies have indicated that oxidative stress and damage are associated with different types of liver disease [8]. With very few treatment options to counter its prevalence, liver cancer remains the world’s third-highest cause of cancer deaths. Natural compounds derived from various sources, such as plants, fruits, vegetables, herbs, and even fungi, open a novel and exciting perspective on the treatment and prevention of hepatic cancer. Terpenoids have emerged as a promising group of phytochemicals amongst the various secondary metabolites produced by plants and other species. Terpenoids selectively kill cells of hepatic cancer with a pleiotropic mode of action while saving normal cells [9].

In this study, UA showed the highest cytotoxicity activity at an IC_50_ value of 0.261 µM (*p* < 0.0001) against the hepatoma cancer cell line (HepG2) compared to its isomer and derivatives, as displayed in Figure 1b and Table 1. Toxins such as CCl_4_ are commonly found in air and water contaminants. This appears to be a strong risk factor when taken in a higher concentration. A cytotoxicity study using HepG2 cell lines has also shown hepatoprotective activity of ursolic acid [7]. Shao et al. [35] investigated the cytotoxicity against HepG2, human cervical carcinoma cells (HeLa), human embryonic lung fibroblast (HELF), human gastric cancer cells (BGC-823), and human neuroblastoma cells (SH-SY5Y). UA (IC_50_ values of 33.1–68.8 µM) showed to inhibit the cell proliferation of all investigated human cancer cell lines [35]. The literature reports that UA (20.6 µM to 65.0 µM) exhibits cytotoxicity activity against HepG2 cancer cell lines [36]. Many researchers have reported plant-based UA to exhibit apoptosis of many human cancer cell lines, including HepG2 [3,37,38,39,40]. Oleanolic acid and ursolic acid, as individual compounds, have biological activities such as hepatoprotection, gastroprotective, anti-tumor, anti-inflammatory, antibacterial, anti-HIV, immunoregulatory, and antihyperlipidemic [35,36,38,39]. The selection of triterpenoids according to an appropriate dose is essential, and this is similar for several herbal ingredients. Triterpenoids have high toxicity under certain circumstances, in addition to the fact that they are relatively safe at low concentrations. OA and UA are reported to act at different phases of tumor development to decrease the initiation of a tumor, promotion, apoptosis, and tumor cell differentiation [41]. Bioassay-guided fractionation using column chromatography has resulted in methyl ursolate isolation with 62.5 μg/mL MIC against *Funtumia africana*. It was moderately toxic to Vero cells and human liver cancer cells with an IC_50_ value of 10.4 μg/mL [42]. Thien et al. [43] reported cytotoxicity activity of 3-O-acetyl ursolic acid against Hep-G2 (IC_50_ = 4.73 µg/mL) [43]. Mngomezulu et al. [14] reported significant dose-dependent cytotoxicity activity of ursolic acid, oleanolic acid, and 3-O-acetyl ursolic acid against HepG2 and HEK293 [14]. Batra and Sastry [44] reported that ursolic acid (*p* < 0.05) and 3-O-acetyl ursolic acid derivative (*p* < 0.01) showed significant homocysteine metabolism and dihydrofolate reductase activity of HepG2 cells [44]. In one study, it was observed that the introduction of acetyl at position C-3 has small positive changes in IC_50_ values against HepG2 [22].

The UA (IC_50_ = 0.00696 µM) (*p* < 0.0001) and MU (IC_50_ = 0.0139) (*p* < 0.001) showed better cytotoxicity activity against PC3 compared to the parent compound UA (IC_50_ = 0.262 µM), which indicates the importance of the addition of the acetyl functional group at position C-3 as shown in Table 1 and Figure 1c. In one research investigation, the lipophilicity induced by the conjugated moiety was essential for the anticancer activity of UA derivatives. IC_50_ values for methyl ursolate derivative were between 6.13 and 11.8 µM in the bladder and pancreatic cell lines [45]. The addition of the acetyl group to the C-3 hydroxyl position of UA or UA derivatives may result in increased cytotoxicity [46]. UA isolated from medicinal plants has many biological activities, such as anti-inflammatory [36], antioxidant [47], anti-carcinogenic [48], anti-diabetes [49], cardioprotective [40], neuroprotective [50], hepatoprotective, anti-skeletal, and thermogenic effects [13]. Current research has confirmed the therapeutic effects of ursolic acid on a wide range of human diseases, particularly different types of inflammation-induced cancer [51]. In one of the investigations, UA showed high cytotoxicity against various cell lines such as human prostate cancer (PC3, DU145, and LNCaP), pancreatic cancer (MIA PaCa-2, PANC-1, and Capac-1), and ovarian carcinoma cells (SKOV-3 and A2780) using the MTT assay [52]. Moreover, Shin et al. [53] stated that UA induces autophagy and apoptosis, including a G1 phase cell cycle arrest in PC3 cells [53]. In in vitro studies, UA has been shown to induce apoptosis in several tumor cells, including PC3 cells, LNCaP cells, human leukemia cancer cells (HL-60), human blood cancer cells (K562), human endometrial adenocarcinoma (HEC108) and human superior cervical ganglion cells (SCG-II), human endometrial cancer cells (EC), human metastatic pigmented malignant melanoma cells (M4Beu), A549 cells, human non-small cell lung carcinoma cells (NSCLC), human Burkitt Daudi cells and aneuploid immortal keratinocyte cells (HaCaT) in a time- and dose-dependent manner [54]. Ursolic acid and its derivatives have been reported to stimulate autophagy in PC3 cells and increase LC3-II expression [51]. OA and UA were documented to exhibit cell proliferation toward breast cancer (MCF-7), colorectal cancer (HT-29), fibroblasts (NIH353), thyroid cancer (8505C), lung cancer (A549), ovarian carcinoma (A2780), WW030272, and melanoma cells (518A2) [52]. Additional research into the effect of ursolic acid (UA) on prostate cancer apoptosis and its possible signal transduction pathway may provide a potential drug target for the clinical treatment of prostate cancer patients [55].

Methyl ursolate exhibited the highest cytotoxicity potency in most of the cell lines, including MDA, HepG2, and PC3, but lower activity on KMST-6 treated cells, which indicates that the addition of methyl at C-28 of UA is essential to enhance its activity as a therapeutic agent for cancer. The 3-O-acetyl ursolic acid showed moderate or no cytotoxicity against the different cancer cell lines, which is less than that of the UA parent compound. This suggests that further modifications of 3-O-acetyl ursolic acid are needed to enhance its activity. Moreover, these results suggest that ursolic acid, its isomer oleanolic acid, and UA derivatives are potential therapeutic drugs for human breast, liver, and prostate cancers.

## 4. Materials and Methods

This investigation reports the isolation of two *Mimusops caffra*-based pentacyclic triterpenic isomers: ursolic acid and oleanolic acid through a synthetic pathway of acetylation and deacetylation. It further discusses the results and how they can be interpreted from the perspective of previous studies and of the working hypotheses. The findings and their implications should be discussed in the broadest context possible. Future research directions are also highlighted.

### 4.1. Collection, Authentication, Isolation, Semi-Synthesis, and Characterization

*Mimusops caffra* was collected from Haga-Haga, Eastern Cape, South Africa, in June 2016. Their authentication and voucher specimen (MS/PL6) deposition were performed at the Botany Herbarium, University of Fort Hare. One kg of dried powdered leaf material from *M. caffra* was subjected to sequential extraction using organic solvents, namely hexane, dichloromethane, ethyl acetate, and methanol. The crude extracts were collected by filtration through Whatman filter paper and solvent removal using a rotary evaporator after 5 consecutive days of shaking twice at 180 rpm with each solvent. Thereafter, the fractions were dried at room temperature to obtain dried crude extracts. Thin Layer Chromatography (TLC) was used to explore the fingerprints of the crude fraction. Ethyl acetate fraction (87.24 g) was used for further isolation of OA and UA. Afterwards, the prominent and better separable fraction was subjected to column chromatography: silica gel 60 (0.063–0.200 mm) to isolate ursolic acid (UA). However, UA (**1**) from *M. caffra* is always accompanied by its isomer oleanolic acid (OA) (**2**) due to their similar retention factors (Rf) values. Therefore, the semi-synthetic experimental methods (acetylation and deacetylation) were used to isolate compounds **1** and **2** with Rf values of 0.760 and 0.771 (hexane: ethyl acetate = 7:3), respectively, in the TLC (Figure 2) [14]. Semi-synthesis (further acetylation and methylation) of ursolic acid was performed to activate some of the functional groups to yield 3-O-acetyl ursolic acid (**3**), methyl ursolate (**4**) and 3-acetylursolic-methylate (**5**) as shown in Figure 3. The isolated ursolic acid and its derivatives were confirmed by TLC (Merck 60F254, Darmstadt, Germany. 0.25 mm), spectroscopic techniques, and melting point [56].

#### 4.1.1. Acetylation of Ursolic Acid and Oleanolic Mixture

The compound **1** and **2** mixture (0.5 g, 1 mmol) was dissolved in pyridine (Merck Gauteng, South Africa) (4.4 mL) with the addition of acetic anhydride (1.0 mL) (pro analysis, Merck Darmstadt, Germany) and 4-dimethylaminopyridine (DMAP) (0.13 g, 1 mmol) with stirring for 24 h at room temperature. Thereafter, the mixture was poured into distilled water with stirring for 1 h. The solid compound was filtered through the suction process, and unreacted DMAP and pyridine were removed by the addition of 2 M hydrochloric acid (HCl). Methanol was used to recrystallize the resulting product, which was air-dried for packing of column chromatography (CC) for isolation of AUA and AOA, using a solvent ratio of n-hexane: ethyl acetate (9:1) [57].

#### 4.1.2. Deacetylation of 3-O-Acetyl Ursolic Acid and 3-O-Acetyl-Oleanolic Acid

The solution was prepared by dissolving AUA (6 mmol) with stirring in methanol (10 mL) with the addition of 10% aqueous potassium carbonate (K_2_CO_3_) (5 mL). In the following step, distilled water was added to the solution, followed by the acidification of the mixture with 5% aqueous HCl and three extractions with ethyl acetate (3 × 40 mL). The organic phase solution was later washed with distilled water (2 × 40 mL). Anhydrous sodium sulphate was used to dry water from the organic layer, followed by the drying of the sample through solvent evaporation, and then recrystallization in methanol [58].

#### 4.1.3. Semi-Synthesis of 3-O-Acetyl Ursolic Acid

A modified method of Basir et al. [59] was used in the preparation of compound **3** (Figure 2). Ursolic acid (0.20 g, 0.44 mmol.) was dissolved in pyridine (5 mL), and excess acetic anhydride (10 mL) was added to a 150 mL round-bottom flask. The mixture was stirred for 12 h at 25 °C for compound **3**. The product was poured into 100 mL of water and stirred for 2 h to hydrolyze excess acetic anhydride. The final product was separated by suction filtration and recrystallized in methanol, then purified by column chromatography [59]. 

#### 4.1.4. Semi-Synthesis of Methyl Ursolate

TBDMS (tert-Butyldimethylsilyl ether) was used to protect the OH of compound 1 at position 3 of UA (**1**), which was dissolved in pyridine (4.4 mL) with the addition of acetic anhydride (1.0 mL) and 4-dimethylaminopyridine (DMAP) (0.13 g, 1 mmol) with stirring for 24 h at room temperature to obtain acetylated AUA. Anhydrous K_2_CO_3_ (0.1 g) and CH_3_I were added dropwise with constant stirring at room temperature to methylate the hydroxyl of the carboxylic acid at position 28. The mixture was constantly stirred for 12 h at 25 °C. The solution was then hydrolyzed with acidified water to remove the acetal protecting group by reversing the protection, through protonation of the oxygen to make a better leaving group.

#### 4.1.5. Semi-Synthesis of 3-Acetylursolic-Methylate

The formation of the methylated ester derivative of ursolic acid (compound **5**) was achieved by a modified method of Wen et al. [60]. AUA (0.2 g, 0.4 mmol) was dissolved in acetone (2 mL), then anhydrous K_2_CO_3_ (0.1 g) and CH_3_I were added dropwise with constant stirring at room temperature. The mixture was constantly stirred for 12 h at 25 °C, then the whole solution was diluted with 100 mL of water and stirred for 2 h. The whole solution was extracted with chloroform, and the organic layer was dried with anhydrous Na_2_SO_4_ and later dried at room temperature, as shown in Figure 2 [60].

### 4.2. Cytotoxicity Activity of Pentacyclic Triterpenes and Derivatives

#### 4.2.1. Cell Culture

The *Mimusops caffra* based pentacyclic triterpenoids and derivatives (ursolic acid (**1**), 3-O-acetyl ursolic acid (**3**), methyl ursolate (**4**), and, 3-acetylursolic-methylate (**5**) were evaluated against various human cancer cell lines such as breast adenocarcinoma cancer cell lines (MDA), human liver cancer cell lines (HepG2), human prostate cancer cell lines (PC3) and non-cancerous human fibroblast cell lines (KMST-6). All the cells used for this study were kindly provided by Prof Mervin Meyer (Department of Biotechnology, University of the Western Cape, Cape Town, South Africa). Human prostate cancer cell lines (PC3) were cultured in Hams F-12 medium containing 1 mM L-glutamine, 5% (*v*/*v*) fetal calf serum and 0.2% (*v*/*v*) streptomycin-penicillin, while breast adenocarcinoma cancer cell lines (MDA), human liver cancer cell lines (HepG2) and non-cancerous human fibroblast (KMST-6) cell lines were cultured in Dulbecco’s Modified Eagle’s Medium (DMEM) medium with GlutaMAX-1, 10% (*v*/*v*) fetal calf serum, and 0.2% (*v*/*v*) streptomycin-penicillin. The cells were allowed to grow to 90% confluency in an incubator set at 37 °C containing 5% CO_2,_ before they were trypsinized and cultured in 96-well tissue culture plates. Stock and final concentrations of the compounds were prepared in the culture media.

#### 4.2.2. Evaluation of Cytotoxicity Using MTT Assay

Cell proliferation was determined using the MTT assay (Sigma-Aldrich, St. Louis, MO, USA (now part of Merck KGaA, Darmstadt, Germany) following the methods described by Wang et al. [35,36,37] with minor modifications. Briefly, the cells were plated in 96-well tissue plates at a density of 2.0 x10^5^ cells per well and treated with various concentrations of the *M. caffra* based pentacyclic triterpenoids and derivatives (ursolic acid (**1**), oleanolic acid (**2**), 3-O-acetyl ursolic acid (**3**), methyl ursolate (**4**), and, 3-acetylursolic-methylate (**5**)), and that ranged from 0.156 µM to 10 µM after which they were incubated for 24 h. Just 5 h before the elapse of 24 h, 10 µL of 5 mg/mL MTT solution was added to each well, and the plates were further incubated. At the end of the incubation period, the media was removed from each well and replaced with 50 µL of DMSO. The plates were shaken on a rotating shaker for 10 min before taking readings at 560 nm using a microplate reader (Elisa Plate Reader, BioTek Instruments, Inc., Winooski, VT, USA). Results of cellular viability were tabulated as the mean absorbance of each compound, expressed as a percentage of the untreated control, and plotted against the compound concentration. IC_50_ values were calculated from the graphs as compound concentrations that reduced the absorbance at 560 nm by 50% of the untreated control wells. To exclude background readings, three wells were seeded with untreated cells in which MTT was not added. Assays were performed in triplicate to ensure reproducibility. The percentage inhibitions of cell proliferation were calculated using the following formula:% Cytotoxicity = ((A − B)/A) × 100
where: A is the absorbance of the negative control (untreated cells). B is the absorbance of the treated cells.

### 4.3. Statistical Analysis

Data were presented as the cumulative mean of each compound. Each compound was run in triplicate. GraphPad Prism (version 5) was used for statistical analysis of all data. For all the data, one-way analysis of variance (ANOVA) followed by Bonferroni’s multiple comparison tests was used to compare the means of one group with every other group. Statistical significance was set at 5%; thus, a *p*-value ≤ 0.05 was considered significant. Concentrations that inhibit cell proliferation by 50% (IC50) were calculated using Graphpad Prism 5 software.

## 5. Conclusions

This study described the synthetic isolation pathway through acetylation and deacetylation of isomers, including ursolic acid and oleanolic acid, with further synthesis of UA derivatives and their basic structure–activity relationship in an attempt to develop new anticancer therapeutic drugs. After evaluating the potency of isolated compounds and derivatives against cancer cells, methyl ursolate showed the highest cytotoxicity potency in most of the cell lines, including MDA, HepG2, PC3, which indicates that the addition of methyl at C-28 of UA is essential to enhance its activity as a therapeutic agent for cancer. The 3-O-acetyl ursolic acid showed moderate or no cytotoxicity against the different cancer cell lines, which is less than that of the UA parent compound. This suggests that further modifications of 3-O-acetyl ursolic acid are needed to enhance its activity. Moreover, these results suggest that ursolic acid, its isomer oleanolic acid, and UA derivatives are potential therapeutic drugs for human breast, liver, and prostate cancers.

## Figures and Tables

**Figure 1 ijms-26-09969-f001:**
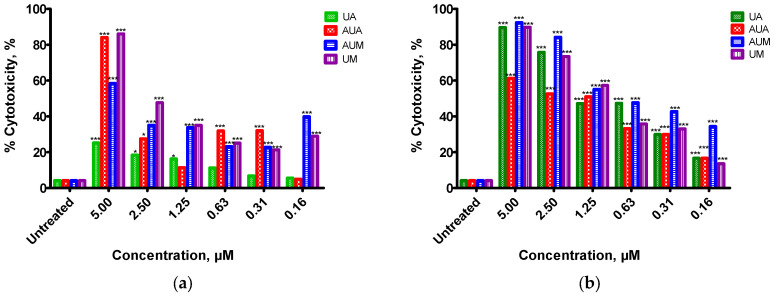
The concentration-dependent cytotoxicity activity of *M. caffra* UA (ursolic acid), AUA (3-acetylursolic acid), AUM (3-acetylursolic-methylate), and UM (ursolic-28-methylate) on (**a**) breast cancer cell line (MDA); (**b**) hepatoma or liver cancer cell lines (HepG2); (**c**) prostate cancer cell line (PC3), and (**d**) human non-cancerous cell line. Each bar represents the cumulative means of each compound in the cell line. Each compound was run in triplicate. * *p* < 0.05, *** *p* < 0.001 statistically compared to untreated.

**Figure 2 ijms-26-09969-f002:**
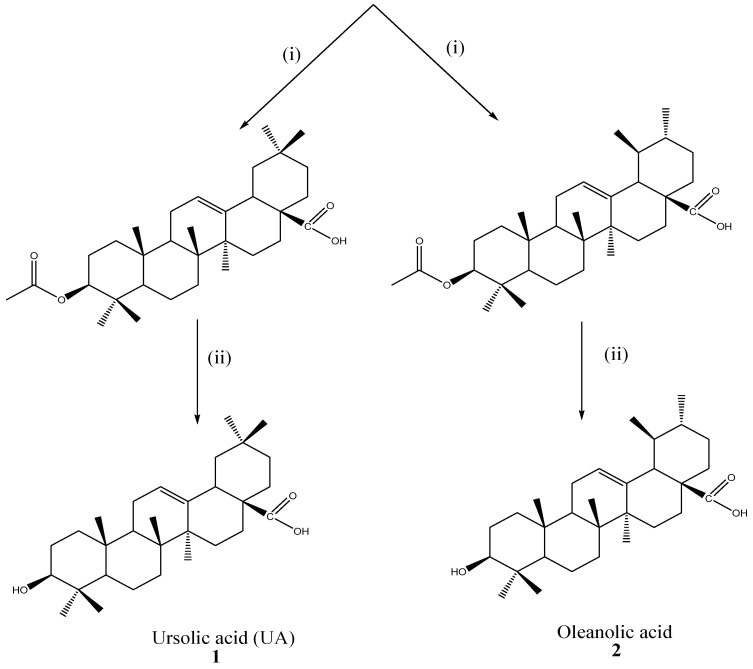
Acetylation and deacetylation of UA/OA mixture, reagents and conditions: (i) Acetic anhydride, pyridine, 4-dimethylaminopyridine (DMAP), 24 h, room temperature; (ii) 10% aqueous potassium carbonate (K_2_CO_3_), methanol (MeOH), 30 min.

**Figure 3 ijms-26-09969-f003:**
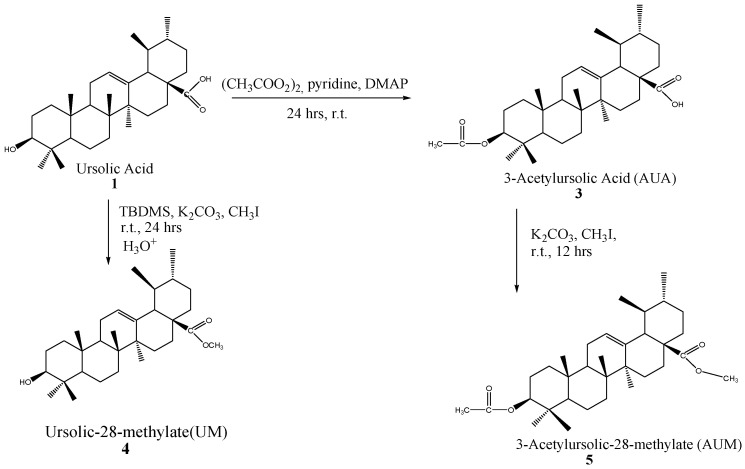
Semi-synthesis of UA to give UM, AUA, and AUM.

**Table 1 ijms-26-09969-t001:** The IC_50_ values for cytotoxicity of isolated compounds and derivatives.

Cells	IC50 Values (µM)				
	1	2	3	4	5
MDA	>5 ^a^	4.48 ^a^	3.93 ^c^	2.48 ^c^	1.90 ^c^
KMST-6	ns	3.88 ^c^	4.88 ^c^	>5 ^c^	>5 ^c^
HepG2	0.261 ^c^	1.20 ^c^	0.850 ^c^	1.60 ^c^	0.850 ^c^
PC3	0.262 ^a^	0.00348 ^c^	ns	0.0139 ^b^	0.0139 ^c^

Graph pad 5 instant: ^a^ = *p* < 0.05, ^b^ = *p* < 0.001 and ^c^ = *p* < 0.0001, ns: not significant.

## Data Availability

Data are contained within the article.

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
