# Peer review of "Cytotoxicity of *Mimusops caffra*-Based Ursolic Acid, Oleanolic Acid and Derivatives Against Human Cancerous and Non-Cancerous Cell Lines"

_ijms, 2025, doi:10.3390/ijms26209969_

Round 1
Reviewer 1 Report
Comments and Suggestions for Authors
This article investigates the cytotoxicity of pentacyclic triterpenoids (ursolic acid, OA) and their derivatives from Mimusops caffra against cancerous and non-cancerous cell lines, with clear focus on structure-activity relationship. The isolation via acetylation/deacetylation, semi-synthesis of derivatives, and MTT-based cytotoxicity assessment are methodologically sound, and key findings (e.g., methyl ursolate’s potent cytotoxicity and cancer cell selectivity) provide valuable insights for anticancer drug development. Minor revisions are needed to enhance clarity and rigor.
- FTIR and NMR spectra should be provided, which can be list as supporting information. This information is very important. At least provide some representative spectra.
- Figure 1, the resolution is too low, please replace with a clearer version. Moreover, the unit of concentration might be wrong, as the unit of IC50 in Table 1 and the concentration of (1)-(5) in 4.2.2 are all μM, while the concentration in Fig.1 is μg/mL.
- How many OA and UA can be extracted from 1 kg of dried powdered leaf material? The extraction efficiency should be added in the main text.
- Line 190, the unit of Mp is missing.
- Section 4.1.4, the method should be provided in detail when the protecting group was removed.
Author Response
Response to Reviewer 1
- FTIR and NMR spectra should be provided, which can be list as supporting information. This information is very important. At least provide some representative spectra.
The FTIR and NMR spectra have been attached as a supplementary file.
- Figure 1, the resolution is too low, please replace with a clearer version. Moreover, the unit of concentration might be wrong, as the unit of IC50 in Table 1 and the concentration of (1)-(5) in 4.2.2 are all μM, while the concentration in Fig.1 is μg/mL.
This was a mistake, and the graphs units has been corrected to μM. The quality – resolution has also been improved and attached as a file.
- How many OA and UA can be extracted from 1 kg of dried powdered leaf material? The extraction efficiency should be added in the main text.
This has been included in the method section and highlighted in yellow.
- Line 190, the unit of Mp is missing.
This has been included and highlighted in yellow.
- Section 4.1.4, the method should be provided in detail when the protecting group was removed.
This has been included as requested.

Reviewer 2 Report
Comments and Suggestions for Authors
The paper discusses new plant-derived substances that can be used to treat breast, liver, and prostate cancers.
Despite the interesting topic, however, I have a few questions and comments for the authors.
1) Besides South Africa, are there any other regions in the world where Mimusops caffra is found?
2) Do the authors have any insight into which part of the plant is most effective for extracting ursolic acid? What would the demand be for these compounds, and thus for the plants, given the potential therapeutic use of such extracts?
3) On what basis were the concentrations of acid and its derivatives selected for MTT analysis to determine cellular viability? Was it based on literature data?
4)The discussion of the results does not include information on how long after exposure of the cells to the compound these MTT results were observed and the IC50 was determined.
5) Why did the authors decide to present the concentrations of the solutions in a different unit than in the graphs in Tables 1 and the discussion? In Table 1 and the discussion, the values are presented in µM, but in the graphs, they are presented in µg/mL. This change in units makes the results and their discussion/comparison illegible. This inconsistency makes the results and their discussion/comparison illegible because it is difficult to read the values on the graphs. I suggest standardizing the concentration units throughout the paper.
6) What do the values "1; 2; 3; 4; 5" under the "IC50 values (µM)" row in Table 1 mean? These values are not explained in the table legend.
7) Line 274: The authors state that the IC50 for UA is 0.261 µM, which can be seen in Graph 1b. What is this concentration value in µg/mL? Should the reader wonder whether they read the value correctly from the small graph? I suggest presenting all values in the same unit from the beginning. Readers encounter similar problems later in the discussion. If the authors provide concentrations in µM in the discussion to facilitate comparison with other data in the literature, I recommend converting the units in the graphs to µM and recalculating the concentrations according to the new unit.
Author Response
Response to Reviewer 2
- Note - the authors did not indicate how many times the experiment was repeated (n=?), what kind of repetitions were these, technical or biological?
The experiment (cytotoxicity) was carried out in triplicate.
2. Note: The authors did not indicate which statistical methods were used and how the data are presented in Figure 1. We see errors and asterisks in the figure, but no explanations.
The statistical method used has been included, and explanations for the asterisks have also been included (highlighted in yellow).
Reviewer 3 Report
Comments and Suggestions for Authors
Cytotoxicity of Mimusops caffra-based ursolic acid, oleanolic acid and derivatives against human cancerous and non–cancerous cell lines
Introduction:
Overall, we see adequate justification for this study, the authors describe the importance of studying specific plant extracts as substances that can exert an antitumor effect.
It would be better if a specific objective of the study was formulated at the end of the section.
Materials and methods:
Section 4.1 is written adequately. Figure 2 reflects the chemical formulas of the substances used.
Section 4.2 describes cell culturing and the method for assessing cytotoxicity. The source of cells and components of the nutrient medium are indicated. The authors used the MTT test, which is a standard solution. The section is written adequately.
Note - the authors did not indicate how many times the experiment was repeated (n=?), what kind of repetitions were these, technical or biological?
Results:
Note: The authors did not indicate which statistical methods were used and how the data are presented in Figure 1. We see errors and asterisks in the figure, but no explanations.
The advantage of this study is that the IC 50 was calculated (Table 1).
Discussion:
The authors discuss the obtained results quite logically and come to the conclusion that ursolic acid, its isomer oleanolic acid and derivatives are potential therapeutic agents for the treatment of breast, liver and prostate cancer in humans.
The abstract corresponds to the content of the article.
Author Response
This work is dedicated to cytotoxic effects of ursolic and oleanolic acid derivatives from Mimusops caffra on cancer and normal cell lines. It was shown that ursolic acid derivatives manifested pronounced cytotoxic effects on breast cancer, hepathoma and prostate cancer cell lines while in general were less active on normal human cells. Despite pronounced biological activity studied in this work, anticancer activity of ursoic acid and its lipophilic derivatives have been extensively studied elsewere as was evidenced by the literature data honestly given by the authors. Therefore, this work lacks novelity and to be published in IJMS, the obtained results need more extensive studies for their interpretation, for example, chemical synthesis of novel analogues, structure-activity relationship and validation of possible targets would be highly advisable.
Thank you for your critical comment, which is highly appreciated. Yes, a lot of work has been done on triterpenes, but it is more on UA. Furthermore, we are reporting both the chemical synthesis analogues and their cytotoxicity to specific cell lines we have in our laboratory.
Reviewer 4 Report
Comments and Suggestions for Authors
This work is dedicated to cytotoxic effects of ursolic and oleanolic acid derivatives from Mimusops caffra on cancer and normal cell lines. It was shown that ursolic acid derivatives manifested pronounced cytotoxic effects on breast cancer, hepathoma and prostate cancer cell lines while in general were less active on normal human cells. Despite pronounced biological activity studied in this work, anticancer activity of ursoic acid and its lipophilic derivatives have been extensively studied elsewere as was evidenced by the literature data honestly given by the authors. Therefore, this work lacks novelity and to be published in IJMS, the obtained results need more extensive studies for their interpretation, for example, chemical synthesis of novel analogues, structure-activity relationship and validation of possible targets would be highly advisable.
Author Response
I appreciate your critical comment and suggestion. I do agree that a lot of work has been done on triterpenes especially OA and BA, but there are little report on UA. Furthermore, we are reporting both the chemical synthesis analogues of what we derivatised and their cytotoxicity to specific cell lines we have in our laboratory.
Round 2
Reviewer 4 Report
Comments and Suggestions for Authors
Despite the authors have added more actual information to their work, it is still being lack of novelity and far from publication in IJMS. As biological properties of ursolic acid are known and extensively studied on various cells lines more additional investigations are needed, at least interpretation of difference in cytotoxicity between normal and cancer cells. On this stage the manuscript can not be accepted.